# Embryology in *Helosis cayennensis* (Balanophoraceae): Structure of Female Flowers, Fruit, Endosperm and Embryo

**DOI:** 10.3390/plants8030074

**Published:** 2019-03-22

**Authors:** Ana Maria Gonzalez, Héctor A. Sato, Brigitte Marazzi

**Affiliations:** 1Facultad de Ciencias Agrarias, Instituto de Botánica del Nordeste (UNNE–CONICET), Sargento Cabral 2131, CP 3400 Corrientes, Argentina; 2Facultad de Ciencias Agrarias (UNJu), Cátedra de Botánica General–Herbario JUA, Alberdi 47, CP 4600 Jujuy, Argentina; hector.a.sato@gmail.com; 3Natural History Museum of Canton Ticino, Viale C. Cattaneo 4, 6900 Lugano, Switzerland; marazzibrigitte@gmail.com

**Keywords:** embryo, endosperm, four-celled embryo sac, holoparasites

## Abstract

*Helosis cayennensis* (Balanophoraceae s.str.) is a holoparasite characterised by aberrant vegetative bodies and tiny, reduced unisexual flowers. Here, we analysed the development of female flowers to elucidate their morpho-anatomy and the historical controversy on embryo sac formation. We also studied the developmental origin of inflorescences and the ontogeny of fruits, embryo and endosperm and discussed in a phylogenetic framework. Inflorescences were analysed by optical, fluorescence and scanning electron microscopy. Inflorescences of *H. cayennensis* arise endogenously. Female flowers lack perianth organs, thus only consist of the ovary, two styles and stigmata. Ovules are undifferentiated; two megaspore mother cells develop inside a nucellar complex. The female gametophyte, named *Helosis*-type, is a bisporic four-celled embryo sac, provided with a typical egg apparatus and a uni-nucleated central cell. Fertilization was not observed, yet a few-celled embryo and cellular endosperm developed. In sum, results confirm that, among Santalales holoparasites, *Helosis* is intermediate in the reduction series of its floral organs. Although perianth absence best supports the Balanophoraceae s.str. clade, our literature survey on female flower developmental data across Balanophoraceae s.l. highlights the many gaps that need to be filled to really understand these features in the light of new phylogenetic relationships.

## 1. Introduction

Balanophoraceae is a relatively small family of 17 genera of root holoparasitic geophytes characterised by an aberrant vegetative and subterranean body, without leaves, stems or roots, called tuber, which may have rhizome-like ramifications [1,2,3,4,5]. These parasitic plants are attached to the root of shrub and tree host species from dark, tropical forests. Inflorescences are the only aerial part of the plant and several of them may appear along rhizomes, making it difficult or impossible to delimit an individual. A peculiarity of the inflorescences is their endogenous origin (in relation to their own tissues), a unique feature in angiosperms [4]. Their flowers are tiny and a wide range of extreme reductions can be observed among genera, especially in female flowers [1,4] and similarly in their seeds, with embryos formed by few cells [2,4]. 

Despite these shared features, recent phylogenetic analyses of Santalales [5] suggested that Balanophoraceae is paraphyletic and divided into two well-supported, unrelated clades: one consisting of the genera *Dactylantus*, *Hachettea and Mystropetalonun*, forming the new family Mystropetalaceae (Clade B), the other including *Helosis* and the remaining genera (*Balanophora*, *Corynaea*, *Ditepalanthus*, *Langsdorffia*, *Lathrophytum*, *Lophophytum*, *Ombrophytum*, *Rhopalocnemis*, *Scybalium*, *Sarcophyte* and *Thonningia*), forming the Balanophoraceae s.str. (Clade A). According to these analyses, no morphological synapomorphies appear to exist to distinguish the two clades but this is probably also due to the lack of knowledge on many aspects of these genera. For instance, with the exception of *Balanophora* and *Dactylanthus*, the mode of germination and establishment of the host/host relationship are unknown [2,4,6] and cultivation of these holoparasites is still not possible, despite years of hard work [7,8]. 

*Helosis* is perhaps one of the relatively less poorly-known genera. It includes three species, the recently discovered *H. antillensis* and *H. cayennensis* (with its two varieties, *var. cayennensis* and var. *mexicana*) and *H. ruficeps* [1,9,10,11]. The present study focuses on *H. cayennensis*, as part of a broader project lead by the first author on the embryology of traditionally circumscribed Balanophoraceae (hereafter referred to as Balanophoraceae s.l.) [12,13,14,15,16]. The species was found in 2006 ([17] fontana), growing in a single spot of about 15 m^2^ in a mesophyll forest of Argentina, where the only evidence of its presence was the appearance of numerous inflorescences at various developmental stages, from barely visible buds hidden in the forest mulch to open mature inflorescences. This finding allowed the beginning of a series of detailed studies on its reproductive anatomy, including the present one.

The highly modified underground vegetative body of *H. cayennensis* is composed of tubers and rhizome-likes branches lacking buds or leaves [2,3,18]. In this monoecious species, 5–10 cm long spadix-like inflorescences emerge in a short period of time, are covered with peltate hexagonal scales when young and unisexual flowers are embedded in a dense layer of filariae [2,4,12,19]. Male and female flowers are clustered, located around each scale of the inflorescence, which acts as a hermaphrodite blossom [1,12]. Gonzalez et al. [12] described the development and structure of the staminate flowers, including pollen formation and the anatomical structure of the inflorescence, whereas female flowers remained unstudied. 

Taxonomic studies describe female flowers of *Helosis* with a concrescence between ovary and calyx and mentioned two alternative perianth forms: an entire ring or bifid papillose projections [1,20,21,22,23,24]. Cardoso and Braga [9] considered the projections at the ovary apex as pieces of perianth and they use this feature in their key of *Helosis* species. However, no ontogenetic studies exist that demonstrate the origin of these projections or that analysed the origin of the filariae covering the inflorescences and their relation to flowers. 

Besides the morphological descriptions in taxonomic studies [1,25,26], the only information about embryo sac and embryo development in *H. cayennensis* is found in Chodat and Bernard [23] (as *H. guyanensis*, a synonym). These authors described the presence of two archesporial cells in a nucellus. During the development of the embryo sac each archesporial cell forms two nuclei, one inferior (denominated antipodal) degenerates and the so-called superior polar nucleus divides twice creating the embryo tetranuclear sac, with one egg cell, a pair of synergids and a central cell. They also described the rapid formation of endosperm and a rudimentary embryo but admit that they were unable to see the fertilization itself. In contrast, Fagerlind [19] reported two megaspore mother cells that undergo meiosis and form a bisporic embryo sac, 8-nucleate and *Allium*-type. He also mentioned that cellular endosperm and a few-celled embryo are produced upon fertilization. Since these two contradictory results, there have been no new studies to re-examine the embryology in this species. This is probably because of the difficulty to collect material, given that it is impossible to detect individuals in the absence of an aerial vegetative body. 

In order to fill gaps in our knowledge of the embryological systems of this root holoparasite and in light of such differences in the descriptions, the purpose of this study aimed at investigating the ontogeny and structure of the inflorescence, female flower, embryo sac, endosperm, embryo and fruit formation of *Helosis cayennensis* var. *cayennensis*. Results were compared with those found in the literature on other holoparasitic species of Balanophoraceae s.l. and discussed in the context of the new phylogeny [5] that separates the family in two unrelated clades.

## 2. Results

### 2.1. Reproductive Phenology

The inflorescence of *Helosis cayennensis* is spadix-like but lacking the spathe typical of a true spadix. The inflorescence has a differentiated axis on a fragile peduncle that raises the oval apical portion with deciduous scales and unisexual flowers embedded in a dense stratum of trichomes called filariae. The male and female flowers are separate but are borne in the same inflorescence. 

The flowering and fruiting period takes place within one month, which occurred in December at the Argentine site. The phases of flowering and fruiting are superimposed on the different inflorescences produced by the same runner. The developmental changes in the inflorescence and flowers were divided in four stages (Figure 1).

Stage I: In the early stages, the inflorescence is underground and develops endogenously from the runners. The axis of inflorescence is covered by a volva that breaks when emerging at ground level (Figure 1a–d). Young inflorescences are covered with tightly arranged scales; peltate, capitate and hexagonal in front view (Figure 1d).Stage II, female phase. All flowers are still covered by scales (Figure 1e–f). Female flowers are the first to develop and open. Anthesis progressed acropetally and the stigmas are exposed on the layer of pink filariae (Figure 1f).Stage III, male phase: When the scales become black and fall off, the male phase begins. The filariae turn light brown, the male flowers expose the anthers above the layer of filariae and pollen grains are released (Figure 1g). By the time when the male flowers are in full anthesis, the styles in the female flowers have mostly fallen.Stage IV, fruiting phase: Inflorescences turn brown to black and the peduncle bends (Figure 1h).

### 2.2. Floral Morphology and Anatomy

The female flowers are in compact groups of 30 to 38 around the base of one scale, the male ones surround them in an outer ring. The average flower density is 74/cm^2^ and the female:male ratio, that is, the floral sex ratio, is 12.6:1. Each female flower is naked, lacking any kind of perianth; it has a gynoecium consisting of a superior ovary and two styles with small capitated stigmata, which are the ones that emerge from the dense layer of filariae (Figure 2a–c). 

The gynoecium reaches a length of 2.24–2.54 mm, of which 1.20–1.26 mm corresponds to the styles (Figure 2c). A short pedicel supports the ovary (Figure 2b), a vascular bundle that derives directly from the inflorescence axis penetrates the pedicel and reaches the base of the ovary, without entering or branching in it (Figure 2e–f). The absence of any cells with tannin in the pedicel allows its differentiation from the ovary (Figure 2b). The ovary wall consists of 3–4 layers of tanniferous cells, without vascular bundles (Figure 2d,f). The entire ovary cavity is occupied by a mass of parenchymatous tissue that is completely fused to the ovary wall, so there is no locule and there is no differentiation of the ovules (Figure 2d). Inside this tissue one (Figure 2d), less frequently two (Figure 2f) embryo sac (ES) develops. A pseudo-endothelium surrounding the ES, their cells contain dense cytoplasm and a large, central nucleus (Figure 2d). The apical portion of the mature ovary has an ovarian crest, it is a ring-shaped excrescence whose cells have a papillose surface (Figure 2d,g,i).

Unlike the ovary, the epidermis in the styles is well defined, consisting of large, vacuolated cells lacking in tannin and covered by a smooth cuticle. The interior region of the styles has 8–12 thin cells in transection and elongated in longitudinal sections (Figure 2g,h,j). The styles do not have any vascularization. The epidermal cells of the stigmata are globose (Figure 2k,l), the cuticle is much distended, because a dense secretion accumulates between the membrane and cuticle, in LS there is a bottle-shaped lumen, conspicuous nucleus and vacuolated cytoplasm. 

Filariae are multicellular trichomes 1.8–2.1 mm long, formed by 3–4 rows of cells; a conspicuous and reticulated cuticle (Figure 2c,j) covers the apical ones. Male flower consists of a 3-lobed tubular perianth and a synandrium composed by three stamens (Figure 2b). 

### 2.3. Inflorescence and Female Flower Ontogeny

In stage I, the young inflorescence, less than 5 mm in height, is still covered by a volva, formed by a few layers of cells originating from runner tissues (Figure 1a,b and Figure 3a). The development of the inflorescence begins with the formation of scales, which are arranged helically on the axis (Figure 1d and Figure 3a,b). 

The scales consist of tanniferous parenchyma, lacking a differentiated epidermis and have a central vascular bundle that derives from the axis of the inflorescence (Figure 3b). The tissue located between the scales is formed by 2–4 strata of compact meristematic cells; there is no clear differentiation into layers (Figure 3b,c). Unlike the scales or the axis of the inflorescence, the meristematic cells lack tannic substances or starch grains in the cytoplasm (Figure 3c).

With the growth of the inflorescence axis, the number of strata in the meristematic zones increases and produce floral primordia (Figure 3d,e). On the rest of the meristematic surface, each cell divides periclinally many times to form compactly arranged filariae (Figure 3d–f). 

The apex of the female flower bud is slightly dome-shaped to flattened (Figure 3d,e). A pair of protrusions develops from the outer edges, which will grow and form two styles and stigmata, which are embedded in the filariae layer at this stage (Figure 3f). The male flower primordium is recognized and distinguished from the female flowers by developing deeper down in the axis of the inflorescence (Figure 3g,h). Within the female floral bud, the apical residue forms a hemispherical structure that grows and occupies the entire ovarian cavity (Figure 3h). In this mass of parenchymatous tissue the placenta and ovules are not differentiated and following Holzapfel (7, see discussion) this structure is called “nucellar complex” (NC), where the embryo sacs (ES) develop in it.

### 2.4. Female Gametophyte Development 

In order to describe the development of the embryo sac and in the absence of an ovule, the term “lower” was used for the base of the NC proximal to the floral pedicel and the term “upper” was used for the distal region of the NC oriented to the styles (Figure 4a).

The NC in the interior of the ovary consists of parenchyma cells that are small and very compactly arranged, with dense cytoplasm. In this parenchymatous tissue, two megaspore mother cells (MMC) are differentiated and they are arranged longitudinally, with 2–4 cell layers separating them from the apex of the NC. The MMCs stand out due to their large size, which reaches 144 × 64 μm, the cytoplasm is clear but very filamentous and a bulky nucleus reaches 49.2–52 μm in diameter (Figure 4a,b). 

In most ovary (up to 96%) one MMC degenerate, leaving necrotic tissue in its place (Figure 4c,e, black arrows). The first meiotic division of the MMC develops a dyad (Figure 4c), the upper megaspore degenerates (Figure 4d–g, arrowheads). The lower remains as functional megaspore (Figure 4d) and originates the binucleate gametophyte (Figure 4d,e, white arrows). A second mitotic cycle gives rise to a tetranucleate gametophyte (Figure 4f–j). With the progressive vacuolation, the four nuclei are pushed towards the periphery of the cell (Figure 4h–j). The following events include both cell growth and increased cytoplasm density, as well as fragmentation of the vacuolar system (Figure 4k). Cytokinesis takes place resulting in an ES composed by four cells: the egg cell have one vacuole oriented towards the apex and two synergids with conspicuous filiform apparatus and vacuoles oriented towards the base of the ES (Figure 4k–m). The central cell have a single nucleus and numerous and small vacuoles (Figure 4l,m). The oosferic apparatus is located towards the upper portion of NC. The pseudo-endothelium remains surrounding the ES, with dense cytoplasmic cells (Figure 4k). Callose is absent both in MMC or during development of ES. In the cases where the two ES have been developed, they have the same ES as previously described. 

### 2.5. Embryo and Endosperm Formation

When gynoecium development is completed and the ES are mature in female flowers, inside anthers of male flowers (of the same inflorescence) microsporogenesis is still in process, indicating the presence of protogyny. However, at the collection site, inflorescences were simultaneously observed in both states: some at female stage with exposed stigmata (stage II) and others at male stage with exposed and dehiscent anthers (state III). Despite the possibility of cross-pollination, the presence of pollen grains on stigmata or pollen tubes within styles or arriving to the embryo sac were not observed at any stage, neither in the analysis with light nor fluorescent microscopy. 

Embryo and endosperm develop before pollen is mature and anthers dehisce in male flowers of the same inflorescence. Even though fertilization was never observed in this study, the egg cell constitutes the zygote and it is located at the upper apex and some small starch grains appear in the cytoplasm (Figure 5a–c). The zygote is quiescent, the nucleus of the central cell divides and the cell undergoes cytokinesis with the formation of an oblique wall in the basal third beginning the endosperm formation (Figure 5b,c). Endospermogenesis continues with successive mitosis followed by cytokinesis (Figure 5d). Due to its development, the pseudo-endothelial cells and nearby NC cells are crushed (Figure 5b–d). 

Only when the endosperm is made of numerous cells, the zygote undergoes a first transverse division (Figure 5e–g). In this state, the male flowers of the same inflorescence are still immersed in the filariae layer; the separation of the tetrads and release of the microspores is observed inside their pollen sacs (Figure 5h). 

In this species, the fruit is the unit of dissemination. The fruit is an achene, 0.5 mm x 1.5 mm long, with a papillose apex. The wall of the fruit (Figure 5h,i) consists of 2–3 outer layers of cells with a completely tanniferous cytoplasm and abundant grains of starch. All fruits collected from completely dried infructescences have undifferentiated embryos, consisting of a few cells and multicellular endosperm. Remains of the pseudo-endothelium with dense cytoplasm surround the endosperm. All analysed achenes, which were already dispersed (and collected in the proximity of the mature and disintegrated infructescence), show an undifferentiated embryo consisting of few cells. No embryos with indications of germination or evidence of adhesion with roots of potential hosts were found. 

## 3. Discussion

Balanophoraceae s.l. is definitely one of the most unusual eudicot families. Indeed, its genera are characterised by aberrant vegetative bodies [4,13,27], highly reduced flowers [2,12,28], inversion and aggressiveness of the ES [15], irregular embryogenesis and small and reduced embryos [2,16,28,29]. This study focused on *H. cayennensis*, a monoecious species with endogenously developed inflorescences. By analysing female flower development and embryology in detail, this study adds to the recent one on inflorescence structure and male flower development [12] and allowed disentangling of the reproductive phenology in this species and of flowering stages (in both male and female flowers) and fruit development. Furthermore, by putting results into a phylogenetic context (Table 1), this study highlights the lack of knowledge in floral development and embryology of holoparasitic Balanophoraceae s.l. taxa. 

### 3.1. Female Flower Development

Reduction is observed at all levels within female flowers of *Helosis* and related genera, starting from the perianth and styles to ovules (Table 1). Developmental studies proved critical to correctly assign organ identity, especially in the case of the perianth, where its identity has long been debated in *Helosis*. Taxonomic studies [1,20,21,22] characterised *H. cayennensis* female flowers as having two perianth forms: one is an entire ring of slightly elongated cells with papillose projections and the other perianth form is bifid, with two inconspicuous segments. Chodat and Bernard [23] described a concrescence between ovary and calyx. Howard [24] reported a bilabiate and triangular perianth in female flowers of *H. cayennensis* (incl. *H. guianensis*) and *H. mexicana*. Cardoso and Braga [9] also considered the entire ring of papillose projections in *H. antillensis* as perianth pieces and used the bifid *versus* entire perianth forms in their key to *Helosis* species. All these previous interpretations were, however, exclusively based on herbarium specimens with no evidence from developmental studies. For instance, our results show that the projections at the apex of the ovary are not perianth pieces, because they are formed at later stages of floral development, after the gynoecium is closed; therefore, they are rather of carpellar origin. Moreover, the two perianth forms actually correspond to different developmental stages: young flowers have two triangular projections between the styles (Figure 2c), whereas they form a ring and occupy the entire apex of the ovary in fully developed flowers (Figure 2i) and in fruits (Figure 5h). Therefore, based on our results, *H. cayennensis* and most likely also the other *Helosis* species do not develop any perianth.

Patterns in perianth presence or reduction appear to be consistent with recent phylogenetic relationships. In genera of Mystropetalaceae [5] a rudimentary perianth occurs in flowers of both sexes (Table 1). We thus agree with Su et al. [5], who state that “*Reductions and losses of floral parts seen in Balanophoraceae* s.str. *(below) are not as pronounced in Mystropetalaceae. For example, a perianth is present on the female flowers of all three genera* [7,28,49]*”*. Mystropetalaceae is sister to the Misodendraceae/Schoepfiaceae clade of hemiparasites [5], in which both families share the presence of flowers with perianth and three undifferentiated ovules (Table 1). 

In contrast, the perianth is generally absent (the gynoecium is naked) or extremely reduced in female flowers of Balanophoraceae s.str. genera [5]. Interestingly, genus *Corynaea*, inferred as sister to *Helosis* ([S5]; formerly together into tribe Helosieae [1]), is taxonomically described as both with and without a perianth and if lacking it, then with two short and broad, lip-like segments instead that protrude above the ovary [1]. We suspect that developmental studies in this genus would reveal that these lip-like segments are in fact not perianth organs but of carpellar origin. Similarly, in female flowers of *Langsdorffia*, *Rhopalocnemis* [5] and *Thonningia*, taxonomic studies [37,38] report the presence of tubular perianths with small, inconspicuous segments fused to the ovary that could also represent a misinterpretation to be clarified with ontogenetic studies. 

Gynoecium development in Balanophoraceae s.l. is also characterised by reductions. Notably, genera display a series of reductions at the ovule level, ranging from ategmic ovules to undifferentiated structures with a diffuse limit between the nucellus and placenta (see Sato [15] and Table 1). Such undifferentiated structures also occur in other Santalales such Schopfiaceae, Misodendraceae and Loranthaceae (Table 1). The lack of clear boundaries between the nucleus and the placenta makes anatomical interpretation of embryological structures difficult in the absence of developmental studies. This probably explains the historical debate in the interpretation of such structures and the subsequent proliferation of terms, in addition to the fact that most of the data in Table 1 are from taxonomic treatments. 

To refer to the area with the diffuse limits, authors who studied the embryology in this family [2,6,7,19,37,50,51,52,53,54,55,56,57,58,59] used a wide array of terms, like: mamelon, nucellar complex, placental-nucellar-complex, papilla, rudimentary ovary, free central placenta and naked (ategmic) orthotropous ovule. The latter three were also used in taxonomic descriptions of *Helosis* [1,19,23]. In our study, however, we preferred to use Holzapfel’s [7] term ‘nucellar complex’ for the central structure in the ovary of *H. cayennensis*, where megaspore mother cells lie and embryo sacs develop. Indeed, Holzapfel [7] writes “*An identification of the central complex as only either nucellus or placenta and consequently the suggestion that one or the other is completely absent*, *may indeed be of little merit at such a strong level of reduction*, *in particular since each term relates in parts to its position relative to the other*” and continued “*However*, *the term ’nucellar complex’ has been chosen here to reflect the absolute necessity of the presence of a nucellus or nucellus remnant in the formation of megaspores*.”

In the hypothetical series of sequential fusion and reductions within the female flowers of Santalales [15], gynoecium ontogeny was studied in only three genera of Balanaphoraceae s.str.: *Balanophora*, *Helosis* and *Lophophytum. Balanophora* is the genus displaying the most reduced gynoecium, as it is a conical and massive body in which one embryo sac is formed [30,31] and a ovarian cavity does not develop [60]. *Lophophytum* is the opposite, as it is the only genus where the ovules are still distinguishable from the placenta; they are ategmic and only consist of a nucellus, without integuments [15]. During gynoecium development in *Lophophytum* two lateral projections develop in the central placenta, resulting into two nucellus primordia, by a post genital fusion the placental-ategmic ovules complex fused with the top of the ovary and thus forms two locules in the mature gynoecium [15]. *Lophophytum* represents the genus with less reduction of gynoecium between Balanaphoraceae s.l. In this hypothetical series of reductions, *Helosis* is intermediate, although among Balanaphoraceae s.l. it possesses the second-most reduced gynoecium. Our study shows that in *H. cayennensis*, after closure of the ovary walls and complete formation of a pair of styles derived from a pair of carpels, the remnant of the floral apex is resolved in a central hemispheric mass or NC with a pair of MMC (Figure 4d). Therefore, we showed that the NC is post genitally fused with the top of the ovary, filling the ovarian cavity. The existence of two MMCs and two styles and stigmata strongly suggests that the ovary of *H. cayennensis* is bicarpellate, a widespread condition in the Balanophoraceae s.l. [1,5,28]; (Table 1). 

### 3.2. Female Gametophyte Development

Female gametophyte development is generally poorly studied in Balanophoraceae s.l. hampering comparison among genera. Furthermore, existing studies might report different results, as is the case of *Helosis*. In the first of the two studies on *Helosis* embryology (Chodat and Bernard [23], who studied *H. guyanensis* = *H. cayennensis*), the megaspore mother cell becomes the megaspore directly and produces two nuclei, one basal or “antipodal” (which soon degenerates) and the second “superior” nucleus that divides and produces an ES with four cells: two synergids, one egg cell and a central one. The present study shows that Chodat and Bernard [23] were partially correct and that the only discrepancy found is in the lower nucleus of the dyad that acts as functional megaspore. In the second study (Fagerlind [19], who studied *H. cayennensis*), two megaspore mother cells in the central papilla (synonym of NC) form a dyad by meiosis; the upper cell degenerates and the lower cell develops an 8-nucleate ES, bisporic and *Allium*-type. In contrast to Fagerlind [19], we never observed an 8-nucleate *Allium*-type embryo, despite having analysed hundreds of gynoecia. Our study indicates that *Helosis* follows a bisporic female gametophyte development, with no wall formation after meiosis II and one mitotic division results in four-celled ES provided with a typical egg apparatus and an enormous uni-nucleated central cell. 

Four-celled, four-nucleated ES, with antipodals absent from the beginning, were described in plants with different pattern of ES formation (Figure 6): in monosporic *Oenothera*-type [47,61] and *Nuphar/Schisandra*-type [characteristics of basal angiosperm taxa: Nymphaeales and Austrobaileyales [62,63,64,65,66]. The *Nuphar*/*Schisandra*-type of ES is similar to the *Oenothera*-type, differs only in the identity of the megaspore in the tetrad that develops into the ES: the chalazal megaspore is functional in the first, whereas the micropylar megaspore in the tetrad is functional in the second type of ES [67]. Similar four-nucleated ES develop from bisporic *Podostemon*-type and *Polypleurum* (=*Dicraea*)-type (Podostemaceae, [68,69] and in tetrasporic *Plumbagella*-type [47]. In bisporic Podostemonaceae and tetrasporic *Plumbagella*-type, the mature ES have diverse cellular configuration, with the four cells arranged in T-shape or in a row [47,70]. The mature embryo sac of *Helosis* is organised in the same way as the *Nuphar*/*Schisandra*-type and *Oenothera*-type; however, it originates from a bisporic development, which prompted us to assign it to its own type, the *Helosis*-type. From a comparative perspective, it is most similar to the *Nuphar*/*Schisandra* type since it derived from the "lower"(chalazal) nucleus (megaspore in the case of *Helosis*). 

The gametophyte of *Helosis* is a variant (with a bisporic pattern of development) that exemplifies the concept of modular construction proposed for the evolution of the angiosperm female gametophyte [71]. According to this hypothesis, an ancestral four-celled/four-nucleate developmental module by duplication led to the seven cells/eight nucleated *Polygonum*-type ES, typical of most angiosperms [64,67,71]. Balanophoraceae s.str. are characterised by a wide variety of ES configurations (Table 1). An embryo sac that is 1-sporic *Polygonum*-type occurs in *Balanophora* [30,31]. Instead, in *Lophophytum* it is 4-sporic *Adoxa*-type [15] but previously described as 1-sporic *Polygonum*-type by Cocucci, [39] and in *Corynaea* it is 2-sporic *Allium*-type [35]. Because the origin and type of embryo sac are fully known only in these four Balanophoraceae s.str. genera [15]; (Table 1), it is not possible to further discuss these features in the context of the new phylogenetic relationships (but see Sato and Gonzalez [15]).

Another characteristic that appears related to the ES of *H. cayennensis* is the presence of a layer of cells with dense cytoplasm that surrounds the ES. This layer is typically found in plants bearing thin unitegmic ovules, where the ES is frequently surrounded by a specialized layer of cells called endothelium or integumentary tapetum, derived from inner cells of the integument [73,74,75,76]. The cells are elongated with dense cytoplasm and become specialized to supply nutrients to the embryo sac. We found such cells of similar location and general morphology in a layer that encircled the MMC and later the ES of *Helosis*. Holzapfel [7] preferred to use “pseudo-endothelium” in flowers *of Dactylanthus taylorii* because they were derived from cells of the nucellar complex, these cells resemble an endothelium of unitegmic ovules. This pseudo-endothelium was observed in *H. cayennensis* also by Chodat and Bernard [23] and Fagerlind [19] and is also reported in some other members of the Balanophoraceae s.l. (Table 1. *Scybalium*: [32]; *Rhopalocnemis*: 2, [37]; *Ditepalanthus*: [19]). 

### 3.3. Endosperm, Embryo and Fruit Development

Fertilization in Balanophoraceae s.l. is another poorly documented and apparently controversial feature. This is not surprising, given the highly reduced structures and the difficulties to identify and interpret them in these taxa. For instance, Umiker [52] and Chodat and Bernard [23] described that fertilization itself does not occur in *Helosis*, the embryo is very rudimentary, composed of a small mass of small cells immersed in the endosperm and devoid of a suspensor. According to Chodat and Bernard [23], the embryo in *Helosis* provides another example of “*nouvel exampled’ apogamie*.” In contrast, Fagerlind [19] described fertilization of ES in *H. cayennensis* and formation of few-celled embryo and a cellular endosperm. Here, we observed no pollen grains on stigmata, pollen tubes inside styles or near ES, neither in analysis with light or fluorescent microscopy. However, we clearly observe the formation of a cellular endosperm and a few-celled embryo, typical of the Balanophoraceae s.l. (Figure 5). This (apparent) absence of fertilization and successful embryo and endosperm formation, leads us to suggest that the embryo formation in *Helosis* occurs by parthenogenesis and that the endosperm development is autonomous. 

Progeny segregation analysis, biochemical and molecular markers or estimation of DNA content or the “egg cell parthenogenetic capacity” tested through the auxin test would be necessary to definitely confirm our observation of parthenogenesis and autonomous endosperm development in *H. cayennensis*. However, during this study it was not possible to collect further plant material for such tests. Furthermore, it is not possible yet to cultivate this species and the tiny flower size does not allow bagging of female flowers or excising of anthers in male flowers. Nevertheless, a similar pattern in embryo and endosperm formation was also described in *Balanophora* [77,78] and *Lophophytum* [16,39], the only other two Balanophoraceae s.str. genera studied ontogenetically. In addition, apomixis was reported by Fagerlind [53] in a few species of *Balanophora*. Further embryological studies on more taxa, coupled with specific tests, are necessary to determine whether this is a shared feature in this newly circumscribed family.

Fruits were described as an achene with few layers of ovary cells contiguous to embryo in *Helosis* and in most genera of studied Balanophoraceae as well [1,15]. The concept of “seed” in the strict sense is, however, not applicable in the genera of Balanophoraceae s.l. [6,7], as there are no ovules, the ESs are embedded in parenchymatic mass devoid of teguments and in many cases it is not possible to even distinguish the nucella from the placenta. Therefore, the unit of dispersion is the achene with an undifferentiated embryo plus endosperm. Actually, *Helosis* and *Balanophora* would be included with in the so-called dust seed plants [79], because their unit of dispersion are among the smallest of all flowering plants. While Martin [80] separates such small-sized seeds into micro (<0.2 mm) and dwarf (0,2-2 mm), Baskin & Baskin [81] simply considers them all as "undifferentiated,” because of the presence of embryos with no cotyledon(s) and radicle. This undifferentiated embryo characterises also several other parasitic plants [2,81,82]. How seeds germinate is, however, known only in a few *Balanophora* species [31]. 

## 4. Materials and Methods 

*Helosis cayennensis* (Sw.) Spreng. var. *cayennensis* was collected in the Apipé Grande Island of the Paraná river, Ituzaingó department, province of Corrientes, Argentina. This collection site is subject to periodic flooding and only two collections could be made: 2009 and 2014, the only couple of years in which the land was not flooded. In the two collections carried out, 83 inflorescences were obtained at different stages of development. According to morphology, the inflorescences were classified into four phenological states: I: flower bud formation, II: female phase: female flowers with exposed stigmata, III: male phase: male flowers with exposed and dehiscent anthers and IV: fruiting.

Material was fixed in FAA (formalin, 70% alcohol and acetic acid, 90:5:5), dehydrated and then embedded in paraffin [83,84]. To carry out the embryological study, three portions of 1 cm^2^ of surface were taken in each inflorescence, at the base, middle and apical region. Each piece was subdivided to make transversal and longitudinal sections (plane in relation to flowers). Serial sections were cut 10 microns thick using a rotative microtome (Microm, Walldorf, Germany). Sections were stained with safranin - fast green [83,84,85]. Lugol was used for identification of starch and FeSO_4_ [83,85] and IKI-H_2_SO_4_ [85] for tannin recognition. Histological sections were analysed using a Leica DMLB2 light microscopy (LM), provided with a LEICA ICC50HD digital camera (Leica Microsystems GmbH, Germany). Polarized light was also used to locate lignified walls and identify the starch grains. 

The sectioned and clarified gynoecium was stained with aniline-blue solution and observed by fluorescence microscopy (Leica DM 1000, Microsystems GmbH, Germany), in order to evaluate the pollen grains on stigmata and pollen tube growth [86,87]. 

Since the flowers are smaller than 3 mm and emerge at different times from the layer of filariae in which they are embedded, density and floral sex ratio (number of male and female flowers) was calculated in 10 inflorescences (stages II and III); the number and sexuality of flower was recorder in 50 fields of 0.5 mm^2^ from each transection of inflorescence (parallel to the surface of the inflorescences). ImageJ software [88] was used for measuring floral and cellular parameters.

For scanning electron microscopy (SEM), the material fixed in FAA was dehydrated in an ascending series of acetone, dried to critical point in CO_2_ (Denton Vacuum LLC, DCP–1, Pleasanton, EUA) and sputter coated with gold–palladium (Denton Vacuum, Desk II, Pleasanton, EUA). SEM observations were performed at the Electron Microscopy Service of the Universidad Nacional del Nordeste (Corrientes, Argentina), using a Jeol LV 5800 microscope (JEOL Ltd., Tokyo, Japan), at 20 Kv. 

Voucher specimens were deposited in the Herbarium of the Botanical Institute of the Northeast (CTES, Gonzalez and Popoff Nº 239; Gonzalez and Sato Nº 470). 

We compiled a list of selected relevant floral characteristics of Balanophoraceae s.l. and of other related families according to recent phylogenetic relationships by Su et al. ([5], Table 1).

## 5. Conclusions

Embryology of holoparasites is a poorly studied subject, perhaps due to the tiny size of their flowers, the complexity of their structure, the difficulty of their collection and the impossibility of cultivating these plants. The present work elucidates previously unclear or controversial aspects on the female embryology in *Helosis* and adds new findings. In particular, it shows that the *Helosis* female gametophyte is bisporic and develops from the lower megaspore cell. The mature embryo sac is composed of a typical egg apparatus and an enormous uni-nucleated central cell. The antipodals never form. The bisporic 4-nucleate/4-celled embryo sac is called *Helosis*-type, because is anatomic and structurally similar to the monosporic 4-nucleate/4-celled *Oenothera*-type embryo sac. No germinating pollen was observed in stigma or growth of pollen tubes in the styles, neither double fecundation, however a few cell embryo develops (typical of the family) and cellular endosperm. By comparing genera of Balanophoraceae s.l. among each other, this study not only supports the position of *Helosis* in the Balanophoraceae s.str. clade (Su [5]) and relationships among them but also clarifies embryological terminology, working towards creating a needed consensus. It also highlights a dramatic lack of knowledge in many aspects of floral development of this fascinating group. As most authors working on holoparasites mentioned, more embryological studies are needed on these species with such extreme reductions. Only with such information, will it possible to find support for current phylogenetic relationships, which suggest a paraphyletic Balanophoraceae s.l. and to increase our understanding of floral evolution of holoparasitic taxa in and beyond this family. 

## Figures and Tables

**Figure 1 plants-08-00074-f001:**
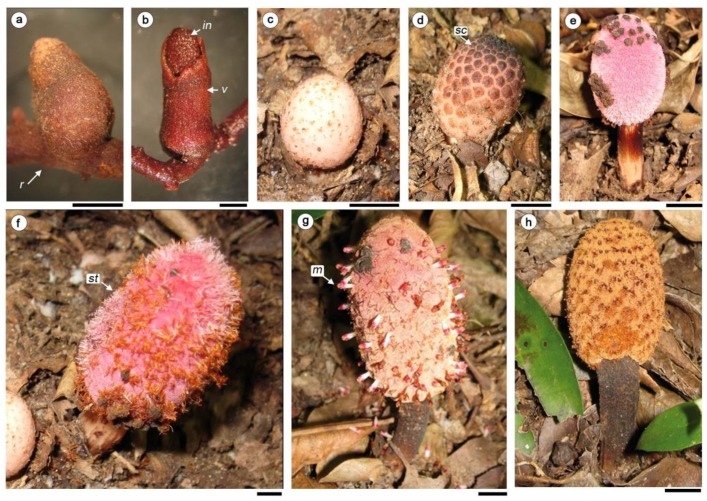
Inflorescence development of *H. cayennensis*, (**a**,**b**) material in FAA, (**f**–**h**) in the field. (**a**,**b**) D stage I. (**a**) Inflorescence covered by volva. (**b**) Volva rupture. (**c**,**d**) Stage II, flowers covered by scales. (**e**,**f**). Stage II, female phase, some scales remain at apex in photo (**e**). (**f**) Stage II, stigmata are exposed over the filariae. (**g**) Stage III: male phase: male flower exposed. (**h**) Stage IV, fruiting phase. Abbreviations: m: male flowers; in: inflorescence, r: runner; sc: scale; st: stigmata; v: volva. Scale bar= (**a**,**b**) 0.2 mm, (**c**–**h**) 1 cm.

**Figure 2 plants-08-00074-f002:**
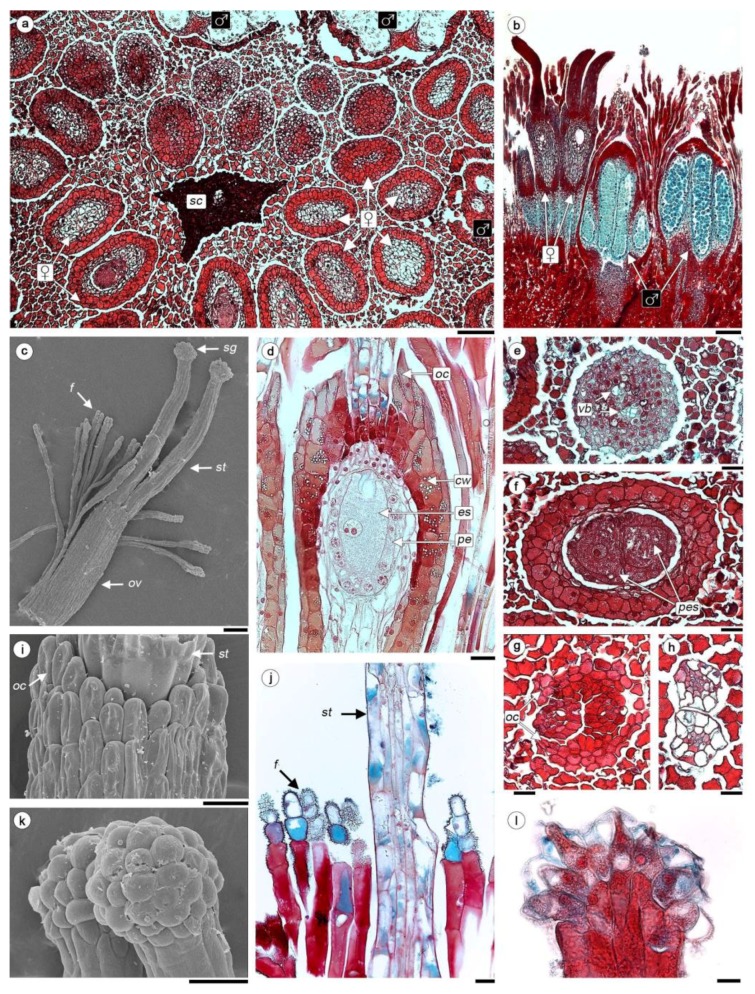
Anatomy of flowers of *H. cayennensis* (LM). (**a**) Paradermal section of inflorescence showing female and male flowers disposition around scale base, all the spaces between flowers are occupied by filariae. (**b**) Longitudinal sections (LS) of inflorescence in the area corresponding to a group of female and male flowers. (**c**) Female flower at embryo sac stage with same filariae attached. (**d**) LS of ovary with mature 4-celled embryo sac and pseudo-endothelium. (**e**–**h**) Serial TS of female flowers, (**e**) Pedicel of flower with vascular bundles, (**f**) ovary with 2 embryo sacs, (**g**) base of styles and ovary crest (**h**) styles. (**i**) Crest at apex of ovary. (**j**) LS with part of style and filariae. (**k**) Stigmata. (**l**) LS of stigmata. Abbreviations: ♀: female flowers; ♂: male flowers; vb: vascular bundles; cw: carpellar wall; es: embryo sac; f: filariae; oc: ovary crest; ov: ovary; pes: pair of embryo sacs; pe: pseudo-endothelium; sc: scale; sg: stigmata; st: style. Scale bar = (**a**–**c**) 0.2 mm, (**c**–**i**) 50 µm, (**j**–**l**) 20 µm.

**Figure 3 plants-08-00074-f003:**
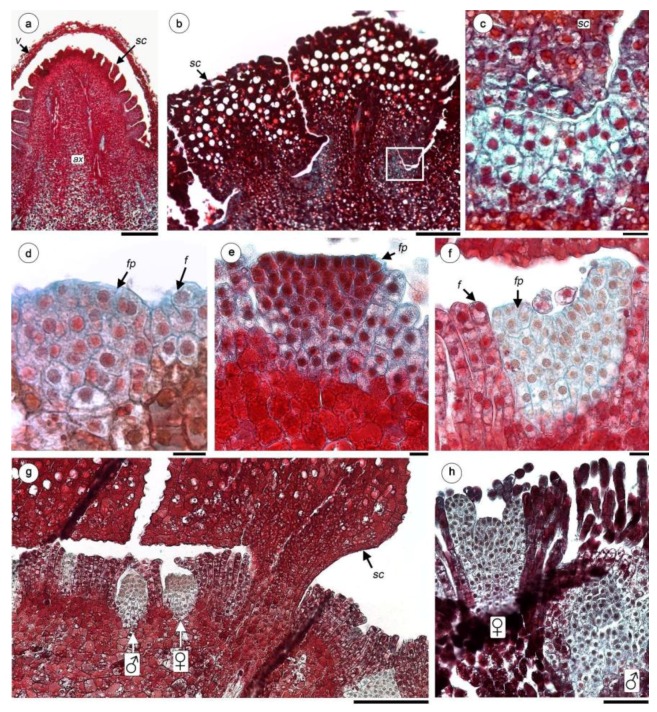
Female flower development of *H. cayennensis* (LM). (**a**) Inflorescence primordium still covered by volva. (**b**) Longitudinal sections of scales—note the meristematic tissue between them (box). (**c**) Detail of meristematic tissue [corresponds to box in (**b**)]. (**d**) Floral and filariae primordium. (**e)** Flattened female floral bud. (**f**) Female floral bud with initiation of styles and stigmata. (**g**) Transection of young inflorescence showing female and male flowers between filariae, all pieces are still covered by scales. (**h**) Young male and female flowers primordium, the gynoecium is still open in the last. Abbreviations: ♀: female flowers; ♂: male flowers; ax: inflorescence axis; f: filariae; fp: female floral primordium; sc: scales; v: volva. Scales bar= (**a**) 0.5 mm; (**b**,**g**,**k**) 200 µm; (**c**–**f**) 20 µm.

**Figure 4 plants-08-00074-f004:**
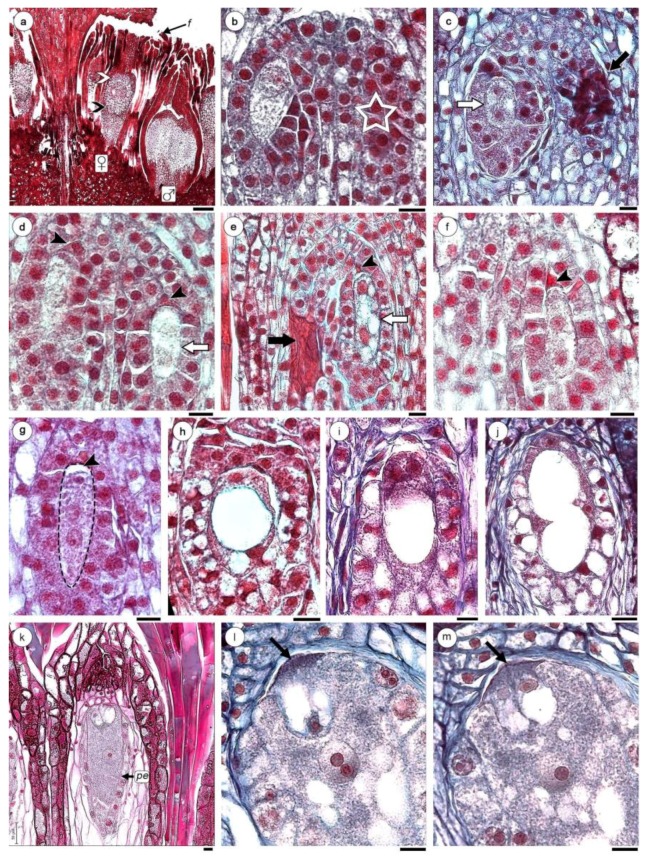
Female gametophyte development of *H. cayennensis* (LM). (**a**) Transection of inflorescence showing male (with microspore mother cells) and female flowers (with MMC), black arrow point indicates “lower” region of the nucellar complex (NC) and white arrow point shows “upper” or distal region of the NC. (**b**) Nucellar complex (NC) with one MMC, the second MMC is not visible in this section, its position is indicated with a star. (**c**) NC with dyad of megaspore (white arrow) and necrotic MMC (black arrow). (**d**) NC with pair of binucleate sporophytes, the apical cell is degenerating (black arrowhead) and basal is in second mitosis (white arrow). (**e**) NC with binucleate and vacuolated sporophytes (white arrow), aborted megaspore (black arrowhead) and necrotic MMC (black arrow). (**f**,**g**) trinucleate stages, black arrowhead indicates megaspore degenerated. (**h**–**j**) Gametophyte with 4 nuclei separated by a central vacuole (only two are visible in the sections **h**,**i**). (**k**) Mature gametophyte showing egg apparatus with two synergids, one egg cell and a central cell containing a single nucleus. (**l**,**m**) Serial sections of oosferic apparatus with filiform apparatus (arrows), (**l**) egg cell and central cell with large nucleus and two nucleoli. (**m**) Longitudinal sections of same embryo sac across synergids. Abbreviations: ♀: female flowers; ♂: male flowers; f: filariae; pe: pseudo–endothelium. Scales bar = (**a**) 0.2 mm; (**b**–**m**) 200 µm.

**Figure 5 plants-08-00074-f005:**
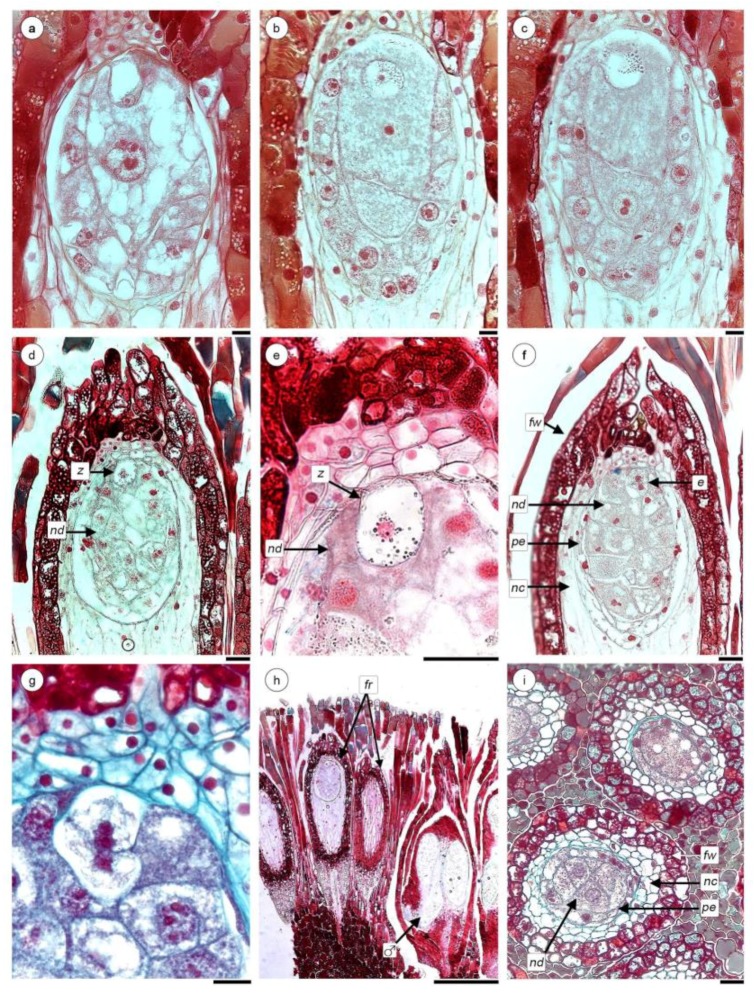
Embryo and endosperm of *H. cayennensis* development. (**a**) Vacuolized embryo sac. (**b**,**c**) Cytokinesis in central cell. (**d**) Longitudinal sections (LS) of fruit with endosperm and zygote. (**e**) Detail of zygote. (**f**) Bicellular embryo and multicellular endosperm. (**g**) Detail of bicellular embryo and surrounded endosperm cells. (**h**) LS of inflorescence with fruit and male flower still not anthetic. (**i**) Transection of fruits. Abbreviations: e: embryo; fr: fruits; fw: fruit wall; nc: nucellar complex; nd: endosperm; pe: pseudo-endothelium; ♂: male flower; z: zygote. Scales bar = (**a**–**c**,**g**) 20 µm; (**d**–**f**,**i**) 50 µm; (**h**) 0.5 mm.

**Figure 6 plants-08-00074-f006:**
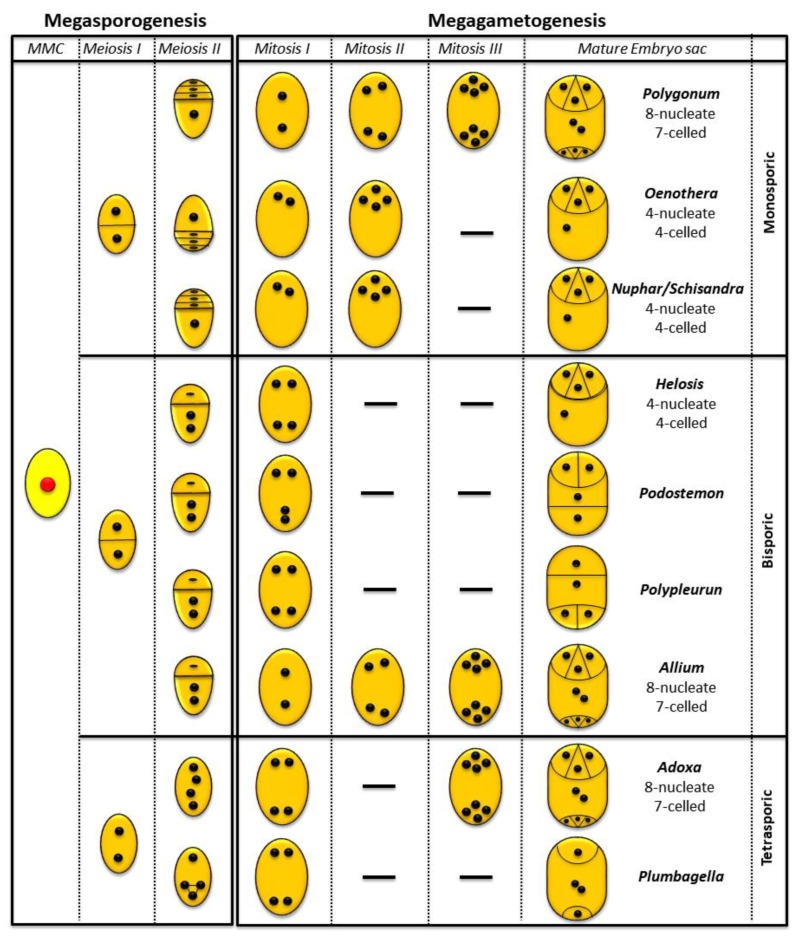
Comparative diagrams illustrating patterns of female gametophyte development between *Helosis* and others types of ES (based on Maheshwari [72], Tobe et al. [65] Johri et al. [6] and Sato & Gonzalez [15]).

**Table 1 plants-08-00074-t001:** Summary of floral characteristics of Balanophoraceae s.l. (Bal. clade A o *sensu stricto* and clade B or Mystropetalaceae) and other related families according to the phylogeny proposed by Su et al. [5].

Family	Genera	perianth (FF) *	styles	locules	ovule/others structures *	MMC/NC	MMC origin	ES type/nº nuclei	PSE	References
Bal. s.str.	*Balanophora* J.R. Forst. & G. Forst.	0	1	0	massive central placental column that later fuses with the ovary wall	1	1, 4-sporic	*Pol/8*	--	[6,28,30,31]
Bal. s.str.	*Langsdorffia* Mart.	tubular inconspicuous segments perianth	1	0	orthotropous, fused to the ovary, placental-nucellar complex ^HS^	1	1-sporic	--	--	[1,22,32,33] ^HS^
Bal. s.str.	*Thonningia* Vahl	0 or tubular inconspicuous segments perianth	1	0	orthotropous, fused to the ovary	--	--	--	--	[34]
Bal. s.str.	*Corynaea* Hook. f.	0 or 2 lip-like segment perianth	2	1	placental-nucellar-complex	2	2-sporic	*Allium/8*	--	[6,35]
Bal. s.str.	*Ditepalanthus* Fagerl.	perianth ring adnate to ovary	2	1	free central placenta	--	--	--	yes	[36]
Bal. s.str.	*Helosis* Rich.	0	2	0 pg	nucellar complex	2	2-sporic	*Helosis/4*	yes	this study
Bal. s.str.	*Rhopalocnemis* Jungh.	2 perianth crests adnate to ovary	2	1	free central placenta, placental-nucellar-complex	2	1-sporic	--	yes	[37,38]
Bal. s.str.	*Scybalium* Schott & Endl.	2 perianth lobes adnate to ovary	2	2 pg	free central placenta, placental-nucellar-complex	--	--	--	yes	[22,32]
Bal. s.str.	*Lathrophytum* Eichler	0	2	2	central placenta	--	--	--	--	[5,22,32,33]
Bal. s.str.	*Lophophytum* Schott & Endl.	0	2	2 pg	placental-nucellar complex, ategmic ovule	2	1-sporic 4-sporic	*Pol/8* *Adoxa/8*	no	[5,15,39]
Bal. s.str.	*Ombrophytum* Poepp. ex Endl.	0	2	2 ^HS^	ategmic ovule, placental-nucellar complex ^HS^	--	--	--	--	[5,22,40] ^HS^
Bal. s.str.	*Chlamydophytum* Mildbr.	0	1	1	--	--	--	--		[41]
Bal. s.str.	*Sarcophyte* Sparrm.	0	1	3	--	--	--	--		[32]
Bal. B. Mystropetalaceae	*Dactylanthus* Hook. f.	2(3) perianth members	1	2 pg	nucellar-complex	2	--	--	--	[7]
Bal. B. Mystropetalaceae	*Hachettea* Baill.	3-lobed cup on top of ovary	1	1	--	--	--	--	--	[4,27,32]
Bal. B. Mystropetalaceae	*Mystropetalon* Harv.	3-lobed cup on top of ovary	1	1	3 ovules arising from a free placenta, each being reduced to an embryo-sac	1	--	*Pol/8*	--	[42,43,44]
Schopfiaceae	*Quinchamalium* Molina, *Arjona* Cav., *Schoepfia* Schreb.	4-5-lobed, epigynous, connate	3	1	3 unitegmic, ategmic ovules (only 1 develops)	1	1-sporic	*Pol/8*	--	[6]
Misodendraceae	*Misodendrum* Banks ex DC.	3-sepaline or vestigial	1	1	3 ovules undifferentiated on a free central placenta	1		*Pol/8*	--	[5,6,27,28,45]
Loranthaceae	*Dendrophthoe* Mart., *Nuytsia* Tiegh., *Struthanthus* Mart.	6-7 petals	1	0-4	ovary-ovule complex, mamelon	1	1-sporic	*Pol/8*	--	[6,28,46,47,48]

Abbreviations = Bal: Balanophoraceae. (FF) *: FF: female flowers, *: terminology used by authors in references. MMC: megaspore mother cell. MMC/NC: number of MMC by nucellar complex or ovules. ES: embryo sac. PSE: pseudo-endothelium presence. Double dashes indicate absence of data (i.e., data unknown). 0: absent, pg: post genital fusion of the placental tip or nucellar complex with the top of the ovary; *Pol*: *Polygonum*. ^HS^: H.A.S personal observations.

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
