# Peer review of "Embryology in Helosis cayennensis (Balanophoraceae): Structure of Female Flowers, Fruit, Endosperm and Embryo"

_plants, 2019, doi:10.3390/plants8030074_

Round 1

Reviewer 1 Report

The author performed comprehensive morphological and anatomic analyses of Helosis cayennensis, a root holoparasitic geophyte belong to the family Balanophoraceae. The author successfully characterized the development of inflorescence, female flower, female gametophyte, embryo and endosperm of Helosis cayennensis. In combination with previous knowledge, this study completed the morphological and embryological survey of this species. Based on these observations and analysis, the author concluded that the embryology of Helosis cayennensis is characteristic of parthenogenesis and autonomous endosperm development.

Although the morphological parts are well done, my major concern is that the manuscript is too descriptive and lacking further evidence to support the conclusion.

The manuscript could be improved in two aspects:

1.       The developmental processes of the inflorescence, female flower, female gametophyte, embryo and endosperm, should be compared to other well-characterized species of Balanophoraceae. The ontogeny of these organs should be further discussed in the context of phylogeny.

2.       Validation of the key molecular markers to support the existence of parthenogenesis and autonomous endosperm development.

Author Response

Comments

The author performed comprehensive morphological and anatomic analyses of Helosis cayennensis, a root holoparasitic geophyte belong to the family Balanophoraceae. The author successfully characterized the development of inflorescence, female flower, female gametophyte, embryo and endosperm of Helosis cayennensis. In combination with previous knowledge, this study completed the morphological and embryological survey of this species. Based on these observations and analysis, the author concluded that the embryology of Helosis cayennensis is characteristic of parthenogenesis and autonomous endosperm development.

Although the morphological parts are well done, my major concern is that the manuscript is too descriptive and lacking further evidence to support the conclusion.

The manuscript could be improved in two aspects:

Point 1. The developmental processes of the inflorescence, female flower, female gametophyte, embryo and endosperm, should be compared to other well-characterized species of Balanophoraceae. The ontogeny of these organs should be further discussed in the context of phylogeny.

 Response 1:

Thank you for this suggestion. To address this request, the first author invited two collaborators to join the paper and contribute to the comparative analyses in a phylogenetic framework, this resulted in addition of table 1 and completely rewritten Introduction and Discussion. Because of  the substantial editing, it was not possible to leave on “track changes”. We hope that by reading the text again, reviewer 1 will appreciate our efforts to address the discussion in a phylogenetic context.

 Point 2: Validation of the key molecular markers to support the existence of parthenogenesis and autonomous endosperm development.

 Response 2:

Thank you. This would indeed be good, but it was unfortunately impossible in our study. First of all because there was no living material available (material was collected in FAA) and a new collection was not possible, because the area does not exist anymore, it was completely altered by recent floodings, and we sadly suspect that the population was lost - unfortunately. Please see the text that we added to address these problems and explain absence of such tests (p. 18 lines 432 to 440, paragraph on “This (apparent) absence of fertilization…”).

Reviewer 2 Report

Look at the attached file!

Author Response

Response to Reviewer 2

We appreciate the work done by the anonymous reviewer. The manuscript has been reviewed and each and every suggestion and correction required by the reviewer has been accepted and corrected.

Each of the reviewer's comments and the corrections made are detailed below:

Comments

Line 4:  Since it is one author there is no need for that

Response: Co-authors have been added, so this line is modified accordingly.

Line 14- all ms :  Avoid the Grammatical & spelling mistakes and personal pronounces (I, we, our etc), use past tense and third person throughout the manuscript. Please apply this to the whole manuscript?

Response: The paper is now by three authors, therefore the use of “we” is justified.

Line 90:  Abbreviations: f: male flowers;

Response: "f" was changed to "m", both in the figure and in the reference.

Line 106:  bv ---- vb: vascular bundles

Response: Corrected

Line 242:  “and references therein

Response: deleted

Lines 267: [19 (pag. 40)]

Response: deleted

Lines 278-283:    “describes”

Response: Corrected to “described”

Line 293:  “and references therein

Response: deleted

Line 301 :  “report”

Response: corrected to “reported”

Lines 307-327:   avoid using short paragraph, build your story in one solid paragraph

Response: Corrected

Line 392:   “cm2”

Response: corrected to cm2

Line 421:   old

Response: deleted

Line 431-433:   Acknowledgments…

Response: Corrected

Reviewer 3 Report

I am not an expert in plant physiology and botany per se. But as far as I read, the manuscript is sound, well written. The data are properly supported with high-quality microscopic pictures. The discussion critically positions the observations with the state-of-the-art. 

The only minor comments I have are included in the enclosed pdf.

Author Response

Response to Reviewer 3

We appreciate the work done by the anonymous reviewer. The manuscript has been reviewed and each and every suggestion and correction required by the reviewer has been accepted and corrected.

Each of the reviewer's comments and the corrections made are detailed below:

Comments

Line 90:   Missing description of g and h

Response: Corrected

Line 106: could be a bit bigger on the panel. Hard to see

Response: In all the panels the size of the internal references was increased. So that they are not confused with the letters that identify each photo,  italics typography was used.

Line 106: vb: where in the panels?

Response: vb were added to the panel

Line 116: “ Inside this tissue one (Fig 2d), less frequently two (Fig 2f) embryo sac (ES) develops” Rev: do you have a number for the frequency?

Response: The correct data is in the next page, where this topic is treated in detail.

Line 123: change 3 by 2

Response: Corrected

Line 137: what is the differences between the three pictures

Response: the references were corrected

Line 146:  “Unlike the scales or the axis of the inflorescence, lack tannic substances or starch grains in the cytoplasm.” What cells? I guess the meristematic cells of the floral primordium. Please specify.

Response: the sentence was corrected

Line 157-158: “...following Holzapfel (19, see discussion) it was decided to call this structure the “nucellar complex” (NC) since one (less frequently two) embryo sac (ES) develops”. Rev: see above remark

Response: The correct data is in the next page, where this topic is treated in detail. The sentence was corrected.

Line 162 : “the term “lower” for the base of the NC proximal to the floral pedicel, and the terms “upper” referring to regions distal of the NC oriented to the styles (Fig. 4a).” Rev: Maybe mark this direction by an arrow in the fig 4a panel?

Response: the arrows were added to the figure and the corresponding reference was corrected

Line 171: “MMC divide by meiosis develop a dyad (Fig. 4c), the upper megaspore degenerates” Rev: 2 verbs in the same sentence

Response: the sentence was corrected.

Line 222-23:  “All fruits collected from completely dried infrutescences have undifferentiated embryos, consisting of a few cells and multicellular endosperm.” Rev: Does it mean the embryo development were never observed beyond 2-cell stage? How does it germinate?

Response:  The mode of germination of this species could not be established from observations made in this study. The sentence was corrected.

In this family the processes of germination, infection, and host/parasite interface are only known in Balanophora where seeds germinate only near a host root (Fagerlind 1948) and endosperm cells form tubular extensions that attach the fruit to the root (Weber and Sunaryo 1990). The embryo of few cells sends the primary haustoria and through a series of divisions produces a nodule (Govindappa and Shivamurthy 1976). information on this lack of knowledge is clarified in the text.

Line 422: typo “confim”

Response: corrected “confirm”.

Line 442: FUNDING INFORMATION (repeated phrase)

Response: deleted.

Round 2

Reviewer 1 Report

The manuscript is substantially improved because of the huge efforts that the authors have invested in the in-depth comparative analysis and discussion in the phylogenetic context. The addition of this part also improved the significance and scientific soundness of the manuscript. 

The authors also provided a reasonable explanation for the absence of further tests to support the existence of parthenogenesis and autonomous endosperm development

Overall, the authors did a great job to improve the manuscript, and the present version is qualified for publication on Plants.